# Does Albumin Predict the Risk of Mortality in Patients with Cardiogenic Shock?

**DOI:** 10.3390/ijms24087375

**Published:** 2023-04-17

**Authors:** Tobias Schupp, Michael Behnes, Jonas Rusnak, Marinela Ruka, Jonas Dudda, Jan Forner, Sascha Egner-Walter, Max Barre, Mohammad Abumayyaleh, Thomas Bertsch, Julian Müller, Ibrahim Akin

**Affiliations:** 1Department of Cardiology, Angiology, Haemostaseology and Medical Intensive Care, University Medical Centre Mannheim, Medical Faculty Mannheim, Heidelberg University, 69117 Heidelberg, Germany; 2European Center for AngioScience (ECAS) and German Center for Cardiovascular Research (DZHK) Partner Site Heidelberg/Mannheim, 68167 Mannheim, Germany; 3Institute of Clinical Chemistry, Laboratory Medicine and Transfusion Medicine, Nuremberg General Hospital, Paracelsus Medical University, 90419 Nuremberg, Germany; 4Clinic for Interventional Electrophysiology, Heart Centre Bad Neustadt, 97616 Bad Neustadt a. d. Saale, Germany; 5Department of Cardiology and Angiology, Philipps-University Marburg, 35037 Marburg, Germany

**Keywords:** cardiogenic shock, albumin, biomarkers, prognosis, mortality

## Abstract

This study investigates the prognostic impact of albumin levels in patients with cardiogenic shock (CS). Intensive care unit (ICU) related mortality in CS patients remains unacceptably high despite improvement concerning the treatment of CS patients. Limited data regarding the prognostic value of albumin in patients with CS is available. All consecutive patients with CS from 2019 to 2021 were included at one institution. Laboratory values were retrieved from the day of disease onset (day 1) and days 2, 3, 4, and 8 thereafter. The prognostic impact of albumin was tested for 30-day all-cause mortality. Moreover, the prognostic performance of albumin decline during ICU treatment was examined. Statistical analyses included univariable *t*-test, Spearman’s correlation, Kaplan–Meier analyses, multivariable mixed analysis of variance (ANOVA), C-Statistics, and Cox proportional regression analyses. In total, 230 CS patients were included, with an overall all-cause mortality at 30 days of 54%. The median albumin on day 1 was 30.0 g/L. Albumin on day 1 was able to discriminate between 30-day survivors and non-survivors (area under the curve (AUC) 0.607; 0.535–0.680; *p* = 0.005). CS patients with albumin < 30.0 g/L were associated with an increased risk of 30-day all-cause mortality (63% vs. 46%; log-rank *p* = 0.016; HR = 1.517; 95% CI 1.063–2.164; *p* = 0.021), which was demonstrated even after multivariable adjustment. Moreover, a decrease of albumin levels by ≥20% from day 1 to day 3 was accompanied by a higher risk of 30-days all-cause mortality (56% vs. 39%; log-rank *p* = 0.036; HR = 1.645; 95% CI 1.014–2.669; *p* = 0.044). Especially when combined with lactate, creatinine, and cardiac troponin I, reliable discrimination of 30-day all-cause mortality was observed, including albumin in CS risk stratification models (AUC = 0.745; 95% CI 0.677–0.814; *p* = 0.001). In conclusion, low baseline albumin levels as well as a decay of albumin levels during the course of ICU treatment, deteriorate prognostic outcomes in CS patients. The additional assessment of albumin levels may further improve risk stratification in CS patients.

## 1. Introduction

Cardiogenic shock (CS) represents the leading cause of death in patients with acute myocardial infarction (AMI) [1,2,3]. CS is characterized by low cardiac output leading to tissue hypoperfusion and hypoxia. CS-related in-hospital mortality rates remain high at about 40–50% despite improved management of CS, including early coronary revascularization, intravenous treatment with vasopressors and inotropes, and the implantation of mechanical circulatory support (MCS) devices [2,4,5,6]. Recently, serum glucose, lactate, and creatinine were identified as useful biomarkers for risk stratification in CS [7,8].

In line with this, with a prevalence of up to 70%, hypoalbuminemia is a common finding in patients in an intensive care unit (ICU) [9]. Albumin is the most abundant plasma protein and a major transporter of blood compounds [10]. In acute and chronic illness, serum albumin may decrease as a result of a transcapillary leak, impaired synthesis related to liver dysfunction, malnutrition, or inflammation [11,12]. Albumin was shown to be a good predictor of short- and long-term outcomes in various settings, such as patients with acute coronary syndrome, acute heart failure, stroke, sepsis, chronic kidney disease, and in older patients [13,14,15,16,17]. Specifically in AMI patients, albumin was shown to increase the risk of all-cause mortality, bleeding events following percutaneous coronary intervention (PCI), and new-onset heart failure [18,19]. 

As a consequence of the overall unacceptable risk of short-term all-cause mortality and the rather poor predictive value of blood-derived biomarkers, improved risk prediction tools are needed to identify individuals at the highest risk of adverse prognosis following CS [20,21]. However, to the authors’ best knowledge, the prognostic impact of albumin on all-cause mortality in patients with CS has not been investigated within adequately powered studies. The authors hypothesize that albumin or the combined predictive value with other biomarkers may improve CS-related risk stratification. Therefore, the present study aims to investigate the prognostic value of albumin on admission, as well as dynamic changes of albumin during ICU hospitalization, on the risk of 30-day all-cause mortality in consecutive patients with CS.

## 2. Results

### 2.1. Study Population

In total, 273 consecutive patients with CS were admitted to our institution from 2019 to 2021. 43 patients with no measurement of albumin on day 1 of ICU treatment were excluded. The final study cohort comprised 230 patients with CS with a median albumin level of 30.0 g/L (IQR 25.5–33.9 g/L). The median age was 74 years, and most patients were males (62%) (Table 1). Comparing 30-day non-survivors and survivors, the rates of cardiovascular risk factors, including arterial hypertension (70% vs. 75%; *p* = 0.342), diabetes mellitus (44% vs. 39%; *p* = 0.397), hyperlipidemia (52% vs. 56%; *p* = 0.525) and smoking (37% vs. 36%; *p* = 0.848) did not differ in both groups (Table 1; middle and right panel). In line with this, the rates of coronary artery disease (38% vs. 37%; *p* = 0.583), congestive heart failure (34% vs. 37%; *p* = 0.665), and atrial fibrillation (33% vs. 34%; *p* = 0.812) were equally distributed. In contrast, especially baseline heart rate was significantly higher in non-survivors (96 bpm vs. 85 bpm; *p* = 0.017). Furthermore, the rates of cardiovascular pharmacotherapies did not differ in both groups.

Table 2 outlines CS-related data and procedures during the course of ICU treatment. AMI was the most common cause of CS in non-survivors (55% vs. 42%), followed by acute decompensated heart failure (ADHF) (27% vs. 23%), whereas CS related to arrhythmias was more common in 30-day survivors (6% vs. 20%) (*p* = 0.002). A minor part of patients presented with CS following pulmonary embolism (*n* = 12; *n* = 4 related to deep vein thrombosis, *n* = 1 patient with antithrombin deficiency), whereas systemic fibrinolysis was performed in *n* = 7 patients, and VA-ECMO in *n* = 1 patient). Meanwhile, three patients revealed CS related to pericardial tamponade (*n* = 1 related to polyserositis; *n* = 2 following coronary perforation after PCI.

Moreover, 30-day non-survivors were admitted with more advanced stages of CS (class E 67% vs. 41%; *p* = 0.001), alongside higher rates of left ventricular ejection fraction (LVEF) < 30% (58% vs. 35%; *p* = 0.003). Furthermore, both the rates of out-of-hospital (OHCA) (45% vs. 32%; *p* = 0.001) and in-hospital cardiac arrest (IHCA) (22% vs. 10%; *p* = 0.001) were higher in the non-survivor group. In line with this, the need for mechanical ventilation (68% vs. 45%; *p* = 0.001) and the norepinephrine doses (median 0.2 µg/kg/min vs. 0.1 µg/kg/min; *p* = 0.001) were higher in non-survivors. Finally, 30-day non-survivors were admitted with lower pH (7.28 vs. 7.32; *p* = 0.003), higher lactate levels (4.5 mmol/l vs. 2.6 mmol/l; *p* = 0.001), higher creatinine (1.60 mg/dl vs. 1.32 mg/dl; *p* = 0.006) and lower albumin levels (28.7 g/L vs. 30.7 g/L; *p* = 0.005) (Table 2).

### 2.2. Correlations with Characteristics and Laboratory Data

Table 3 displays the univariable correlations of albumin with clinical and laboratory data. Albumin on day 1 inversely correlated with age (*r* = −0.147; *p* 0.026). This was in line with a correlation with C-reactive protein (CPR) (*r* = −0.433; *p* 0.001), procalcitonin (*r* = −0.392; *p* 0.001), N-terminal pro-B-type natriuretic peptide (NT pro-BNP) (*r* = −0.596; *p* 0.001) and creatinine (*r* = −0.285; *p* 0.001). Furthermore, albumin correlated with important ICU scores such as the sepsis-related organ failure assessment (SOFA) (*r* = −0.409; *p* 0.001), acute physiology (*r* = −0.339; *p* 0.001), and the acute physiology and chronic health evaluation II (APACHE II) scores (*r* = −0.413; *p* 0.001). Finally, albumin was inversely correlated with fluid balance (*r* = −0.201; *p* 0.003).

### 2.3. Prognostic Impact of Albumin Levels

In patients admitted with CS, the overall rate of 30-day all-cause mortality was 54%. During the first week of ICU treatment, albumin levels were significantly lower in 30-day non-survivors compared to survivors on day 1 (28.7 g/L vs. 30.7 g/L; *p* = 0.005), day 2 (26.1 g/L vs. 28.0 g/L; *p* = 0.002), day 3 (23.5 g/L vs. 26.6 g/L; *p* = 0.001), day 4 (22.3 g/L vs. 25.1 g/L; *p* = 0.001), and on day 8, respectively (20.5 g/L vs. 24.1 g/L; *p* = 0.001) (Figure 1). Regarding C-statistics, albumin displayed moderate discrimination for 30-day all-cause mortality during the first week of CS treatment in consecutive CS patients (AUC 0.607–0.753; *p* ≤ 0.005) (Table 4).

The mixed ANOVA showed that survival alone had a statistically significant effect on albumin levels during the first 8 days of ICU treatment (*p* = 0.007). Similarly, time alone significantly affected albumin levels (*p* = 0.001). The interaction of time with survival was not detected (*p* = 0.645) (Figure 2).

At 30 days, the primary endpoint all-cause mortality occurred in 63% of patients with albumin below the median (i.e., <30 g/L) and in 46% of patients with albumin above the median (log-rank *p* = 0.016). Accordingly, low albumin was associated with an increased risk of 30-day all-cause mortality (HR = 1.517; 95% CI 1.063–2.164; *p* = 0.021) (Figure 3; left panel). Increased risk of all-cause mortality was specifically observed in patients with AMI-related CS (74% vs. 49%; log-rank *p* = 0.006; HR = 1.866; 95% CI 1.154–3.016; *p* = 0.011), whereas albumin was not associated with the risk of all-cause mortality in patients with non-AMI related CS (56% vs. 40%; log-rank *p* = 0.111; HR = 1.517; 95% CI 0.893–2.576; *p* = 0.123) (Figure 3; middle and right panel).

Even after multivariable adjustment with baseline characteristics, clinical and laboratory data, albumin < 30.0 g/L on day 1 was still associated with increased risk of the primary endpoint (HR = 1.875; 95% CI 1.109–3.170; *p* = 0.019) (Table 5). Even when included as a continuous variable, albumin was associated with the primary endpoint (HR = 0.917; 95% CI 0.864–0.974; *p* = 0.005). 

Taking a further in-depth analysis within pre-specified subgroups, the risk of 30-day all-cause mortality was not increased in patients with hypalbuminemia in the subgroups of patients with BMI < 25 kg/m^2^ (HR = 1.125; 95% CI 0.516–2.456; *p* = 0.767), patients with more than 80 years of age (HR = 1.353; 95% CI 0.475–3.855; *p* = 0.571), patients with malignancies (HR = 1.017; 95% CI 0.519–2.106; *p* = 0.812), and in patients with multi-organ failure (HR = 1.423; 95% CI 0.845–2.398; *p* = 0.185). This suggests that a pre-selected subgroup did not trigger the prognostic value of albumin levels.

### 2.4. Prognostic Impact of Albumin Decrease during Course of ICU Treatment

Thereafter, the prognostic role of albumin decrease from day 1 to day 3 was investigated. Patients with albumin decrease ≥ 20% from day 1 to day 3 were associated with increased risk of 30-day all-cause mortality (56% vs. 39%; log-rank *p* = 0.036; HR = 1.645; 95% CI 1.014–2.669; *p* = 0.044) (Figure 4). Finally, a decrease in albumin was still significantly associated with an increased risk of the primary endpoint after multivariable adjustment (HR = 1.737; 95% CI 1.029–2.933; *p* = 0.039) (Table 6).

### 2.5. Prognostic Impact of Albumin Supplementation

In hypalbuminemic patients, albumin supplementation was performed in 20% (median 80 g during ICU hospitalization). However, albumin supplementation was not associated with the risk of 30-day all-cause mortality in CS patients (74% vs. 60%; log-rank *p* = 0.729; HR = 1.095; 95% CI 0.635–1.889; *p* = 0.743) (Figure 5).

### 2.6. Combined Use of Biomarkers for Risk Prediction in CS Patients

Within the entire study cohort, established biomarkers, such as creatinine (AUC = 0.605; *p* = 0.005), lactate (AUC = 0.677; *p* = 0.001), as well as cardiac troponin I (AUC = 0.675; *p* = 0.001) were associated with moderate predictive value for 30-day all-cause mortality (Figure 6). However, when combined with albumin, risk stratification with regard to 30-day all-cause mortality was improved with an AUC of 0.745 (95% CI 0.677–0.814; *p* = 0.001) within the combined ROC analysis.

## 3. Discussion

The present study sought to investigate the prognostic value of albumin in consecutive CS patients admitted to an internistic ICU. The data suggests that albumin levels were significantly lower in 30-day non-survivors compared to survivors during the first week of ICU treatment. Albumin on day 1 showed reliable discrimination for 30-day all-cause mortality (AUC 0.607; *p* = 0.005). In addition, albumin < 30.0 g/L was associated with an increased risk of 30-day all-cause mortality, which was consistent after multivariable adjustment. Furthermore, a decrease of albumin by ≥20% from day 1 to day 3 of ICU treatment was specifically associated with impaired prognosis. Finally, the combined assessment of creatinine, lactate, cardiac troponin I, and albumin was associated with superior discrimination of 30-day all-cause mortality compared to isolated biomarker measurement. Therefore, the additional assessment of albumin levels may further contribute to an improved risk prediction in CS patients.

Until now, the prognostic value of low albumin levels has been investigated within various studies, including patients with AMI, suggesting an increased risk of all-cause mortality in patients admitted with low albumin levels. In contrast, no further sub-analyses were performed by the presence or absence of concomitant CS [19,22]. Furthermore, Islam et al. demonstrated adverse in-hospital outcomes, including higher rates of heart failure, arrhythmias, and CS in patients with albumin < 35 g/L, including 374 patients with the first episode of AMI [23,24]. In line with this, higher rates of heart-failure-related hospitalization and cardiovascular death were demonstrated in patients with low albumin irrespective of severity and subtype of AMI in 2253 AMI patients during a median follow-up of 3.2 years [25].

The albumin concentration is affected by various conditions, including endogenous synthesis, distribution, exogenous albumin loss, and hemodilution. Similarly, albumin synthesis may be impaired by concomitant liver disease, nutrition status, infection, and specifically in sepsis [26,27,28,29]. In patients suffering from AMI, adverse outcomes of patients with low albumin levels may be attributed to higher rates of older patients with consecutively increased rates of malnutrition and cachexia, reflected by lower cholesterol and lymphocyte levels in prior studies [22]. However, in the setting of CS, mechanisms that may contribute to the adverse prognosis of patients with lower albumin levels may include increased bleeding rates related to coronary revascularization and impaired cardiac output by the interaction of albumin and cardiac myocytes [11,30]. However, within the present study, the increased risk of 30-day all-cause mortality in patients with hypalbuminaemia was not trigged by subgroups of patients with very low BMI or concomitant malignancies (i.e., in patients with increased risk of malnutrition), suggesting an independent association of albumin with the risk of 30-day all-cause mortality in the setting of CS.

Although AMI-related CS rates were demonstrated to decrease within the last few years, AMI still represents the most common cause of CS [1,3]. Within the present registry, 49% of the patients developed CS following AMI, predominantly in the non-survivor group (55%). In patients with AMI-related CS, low albumin may increase the risk of bleeding events. For instance, an increased risk of bleeding events at 3 years in patients with hypalbuminaemia (<38 g/L) was observed by Yoshioka et al. Their study included 1724 patients undergoing PCI following AMI [18]. Increased risk of major bleeding in patients with low albumin levels may be attributed to vitamin C and K deficiency, specifically in the setting of malnutrition [31]. With regard to the interaction of albumin and cardiac function, registry data suggested lower LVEF in patients with hypoalbuminemia [22]. In those patients, LVEF may be impaired by the lack of anti-inflammatory and antioxidant effects in patients with low albumin levels [30,32]. However, no association between albumin and baseline LVEF was found within the present registry. The inclusion of blood-derived biomarkers in risk stratification patients with AMI-related CS may be of major importance since the identification of appropriate risk prediction tools following AMI-related CS remains challenging. Good myocardial reperfusion following PCI and renal function, assessed with glomerular filtration rate (GFR), was recently demonstrated to improve outcomes in AMI-related CS patients [33]. The prognostic value of the renal function in CS was verified within a sub-study of the CULPRIT-SHOCK study [34]. Since limited data in CS patients exists, suggesting lactate and creatinine/GFR are predictive for 30-day all-cause mortality, the additional value of albumin was tested for 30-day all-cause mortality [35]. It was demonstrated that adding albumin to risk prediction models in line with established biomarkers, such as lactate, creatinine, and cardiac troponin I, improve overall risk prediction in CS patients.

The above-mentioned studies were not restricted to patients treated at an ICU; only a minor part of AMI patients with concomitant CS was included. Low albumin (i.e., <3.5 g/dL) was shown to increase the risk of in-hospital and post-discharge mortality among 2680 patients in a cardiac ICU. However, most patients were admitted with heart failure, followed by acute coronary syndrome, whereas less than 20% of patients suffered from concomitant CS [9]. Until now, the prognostic value of albumin levels has been investigated within one study, which included 178 CS patients. Thus, Jäntti et al. demonstrated hypoalbuminemia on admission was associated with an increased risk of 90-day all-cause mortality, which was still observed after adjustment for established CS scores (i.e., CardShock risk score or IABP II shock score). However, a decline in albumin levels was not associated with the risk of all-cause mortality in their study [36]. Within the present study, including consecutive CS patients, albumin was demonstrated to be an independent predictor of all-cause mortality, demonstrated during ICU treatment following the first week of ICU admission. Interestingly, a decrease in albumin during the course of ICU treatment was associated with an increased risk of 30-day all-cause mortality. In contrast to the findings by Jäntti et al., follow-up albumin measurements were performed during the first week of CS (i.e., on day 1 until day 8), whereas albumin measurements beyond 72 h of admission were beyond the scope of their study.

Apart from studies investigating the prognostic role of albumin in CS, Peng et al. recently investigated the prognostic role of the ratio of neutrophils to albumin (i.e., neutrophil-albumin-ratio (NAR)) since inflammation contributes to CS progression in many patients [37]. Including 475 patients with CS, the authors demonstrated that the NAR revealed improved prognostic accuracy compared to neutrophils or albumin alone. However, the inclusion of other blood-derived biomarkers was beyond the scope of their study. However, suggesting more than half of the patients with CS within the present study suffered from infection supports the importance of investigating the NAR in patients with CS further.

The present study’s findings raise the question of whether albumin supplementation may be beneficial in patients with CS. The combination of crystalloid fluid administration and albumin supplementation was investigated by Zou et al., including 1586 CS patients from the MIMIC-IV database. It was demonstrated that an early albumin supplementation was associated with decreased risk of 30- and 60-day all-cause mortality even after propensity score matching. However, the presence of pulmonary edema and distinct volume status was beyond the scope of their study [38]. Within the present study, albumin supplementation was performed in 20% of CS patients with concomitant hypalbuminaemia, whereas no prognostic benefit of albumin supplementation was suggested. Whether an early albumin administration may reduce the risk of all-cause mortality in CS patients must be determined in prospective, randomized trials.

In conclusion, the present study suggests an increased risk of 30-day all-cause mortality in CS patients with low albumin (i.e., <30 g/L), which was consistent after multivariable adjustment. Furthermore, a decrease of albumin by >20% from day 1 to day 3 during the course of ICU treatment was associated with increased 30-day mortality. However, albumin supplementation was not associated with outcomes in the present registry. Finally, the additional assessment of albumin was associated with superior discrimination of 30-day all-cause mortality compared to when analyzed in combination with lactate, creatinine, and cardiac troponin I. This suggests that albumin measurement may further improve CS patients’ risk prediction.

This study has several limitations. Due to the single-center and observational study design, results may be influenced by measured and unmeasured confounding. Liver function tests were not performed within the present study. As a result of a rather low frequency of neutrophil measurement within the present study, the NAR was beyond the scope of the present study. In patients admitted with AMI-CS, no information on pain-to-balloon times was available for the present study. Finally, no information on long-term mortality was available for the present study.

## 4. Materials and Methods

### 4.1. Study Patients, Design, and Data Collection 

The present study prospectively included all consecutive patients presenting with CS on admission to the internistic ICU at the University Medical Center Mannheim, Germany, from June 2019 to May 2021. All relevant clinical data related to the index event were documented using the electronic hospital information system as well as the IntelliSpace Critical Care and anesthesia information system (ICCA, Philips, Philips GmbH Market DACH, Hamburg, Germany) implemented at the ICU, organizing patient data such as admission documents, vital signs, laboratory values, treatment data, and consult notes. 

Important laboratory data, ICU-related scores, hemodynamic measurements, and ventilation parameters were assessed on the day of admission (i.e., day 1) and on days 2, 3, 4, and 8. Furthermore, baseline characteristics, prior medical history, length of index hospital stay, data derived from imaging diagnostics, and pharmacological therapies were documented. Documentation of source data was performed by intensivists and ICU nurses during routine clinical care. Source data was reviewed for plausibility by two independent cardiologists blinded to the final data analyses.

The present study derived from an analysis of the “Cardiogenic Shock Registry Mannheim” (CARESMA-registry), representing a prospective single-center registry including consecutive patients presenting with cardiogenic shock being acutely admitted to the ICU for internal medicine of the University Medical Center Mannheim (UMM), Germany (clinicaltrials.gov identifier: NCT05575856). The registry was carried out according to the principles of the Declaration of Helsinki and was approved by the medical ethics committee II of the Medical Faculty Mannheim, University of Heidelberg, Germany.

The medical center covers a general emergency department (ED) for emergency admission of traumatic, surgical, neurological, and cardiovascular conditions. Interdisciplinary consultation is an inbuilt feature of this 24/7 service and connects to a stroke unit, four ICUs, and a chest pain unit (CPU) to alleviate the rapid triage of patients. The cardiologic department includes a 24 h catheterization laboratory, an electrophysiologic laboratory, a hybrid operating room, and telemetry units. Since 2020, the University Medical Center has been a certified cardiac arrest center (CAC), including the ability to implant extracorporeal life support (ECLS) devices, such as Impella and veno-arterious extracorporal membrane-oxygenation (VA-ECMO) (i.e., i-cor^®^, Xenios AG, Heilbronn, Germany and Cardiohelp, Getinge, Gothenburg, Sweden) [39,40,41].

### 4.2. Inclusion and Exclusion Criteria, Study Endpoints

For the present study, all consecutive patients with CS treated at one internistic ICU with measurement of albumin on day 1 were included. No further exclusion criteria were applied. The diagnosis of CS was determined according to the current recommendations of the Acute Cardiovascular Care Association of the European Society of Cardiology [42]. Accordingly, cardiogenic shock was defined by hypotension (systolic blood pressure (SBP) < 90 mmHg) for more than 30 min despite adequate filling status or need for vasopressor or inotropic therapy to achieve SBP > 90 mmHg. Additionally, signs of end-organ hypoperfusion must be present such as oliguria with urine output < 30 mL/hour, altered mental status, cold, clammy skin, and increased lactate > 2 mmol/L. All-cause mortality at 30 days was documented using the electronic hospital information system and by directly contacting state resident registration offices (‘bureau of mortality statistics’). Identification of patients was verified by place of name, surname, day of birth, and registered living address. No patient was lost to follow-up with regard to all-cause mortality at 30 days. 

### 4.3. Measurement of Albumin Levels

After adequate clotting of serum samples for one hour at room temperature, samples were centrifuged at 2000× *g* for 10 min at 18 °C, according to the manufacturer’s recommendation. Albumin measurements were performed fully automatically by extinction measurement based on the color binding method, according to Carter and Louderback. The reference range for albumin in adults has been reported to be 34–50 g/L. The detection limit (LoD) of the method is reported by the manufacturer as 6 g/L, and the assay has a linearity range of 5 to 80 g/L. All investigations were carried out in an accredited laboratory under DIN-EN ISO 15189 conditions.

### 4.4. Statistical Methods

Quantitative data were presented as mean ± standard error of the mean (SEM), median and interquartile range (IQR), and ranges depending on the distribution of the data. They were compared using the Student’s *t*-test for normally distributed data or the Mann–Whitney U test for nonparametric data. Deviations from a Gaussian distribution were tested by the Kolmogorov–Smirnov test. Qualitative data were presented as absolute and relative frequencies and were compared using the Chi-square test or Fisher’s exact test, as appropriate. Box plots for the distribution of albumin were created for the comparisons of survivors and non-survivors on days 1, 2, 3, 4, and 8. Spearman’s rank correlation for nonparametric data was used to test for the association of albumin with medical and laboratory parameters on day 1. C-statistics were applied by calculating the receiver operating characteristic (ROC) curves and investigating the corresponding areas under the curves (AUC) within the entire cohort to assess the diagnostic performance of albumin during the first week of ICU hospitalization with regard to the 30-day all-cause mortality. The albumin levels over time (i.e., days 1, 2, 3, 4, and 8) were analyzed using mixed factorial analysis of variance (ANOVA) to estimate the effects of the two factors, time and survival, on biomarker levels. The sphericity was tested using the Mauchly test, and a Huynh–Feldt correction was applied to the mixed ANOVA results in case of not fulfilling the spherical assumption. The Huynh-Feldt correction was preferred over the Greenhouse–Geisser correction because the latter tends to increase the risk of type II error by being more conservative [43]. Kaplan–Meier analyses according to albumin were performed, and univariable hazard ratios (HR) were given together with 95% confidence intervals within the entire study cohort and separated by AMI and non-AMI related CS. Thereafter, the prognostic role of an albumin decrease by ≥20% from day 1 to day 3 of ICU treatment was assessed compared to patients with no albumin decrease or album decrease < 20%. The prognostic impact of albumin was confirmed using multivariable Cox regression models using the “forward selection” option. Finally, C-statistics were applied by calculating the ROC curves and investigating the corresponding AUC within the entire cohort, to assess the diagnostic performance of albumin levels as compared to other biomarkers (i.e., creatinine, lactate, cardiac troponin I) with regard to the 30-day all-cause mortality. Furthermore, combined C-statistics (i.e., lactate, cardiac troponin I, albumin, creatinine) were performed using multivariable logistic regression analyses.

Results of all statistical tests were considered significant for *p* ≤ 0.05. SPSS (Version 28, IBM, Armonk, NY, USA) and GraphPad Prism (Version 9, GraphPad Software, San Diego, CA, USA) were used for statistics. 

## Figures and Tables

**Figure 1 ijms-24-07375-f001:**
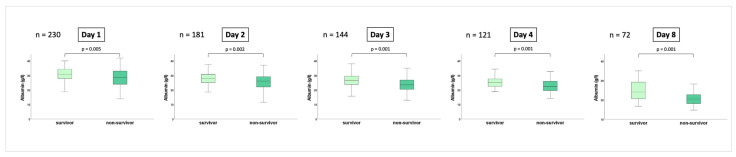
Box plots demonstrating the distribution of albumin among 30-day survivors and non-survivors with CS during the first 8 days of CS onset (i.e., on days 1, 2, 3, 4, and 8). The data are presented as the median with interquartile ranges (boxes) and 5–95% percentiles (whiskers).

**Figure 2 ijms-24-07375-f002:**
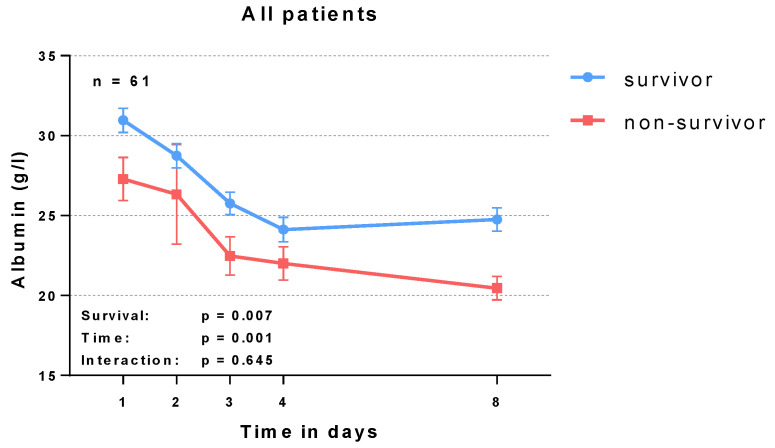
Mixed analysis of variance (ANOVA) for albumin dynamics during the first 8 days of SC onset according to survivors and non-survivors at 30 days of follow-up.

**Figure 3 ijms-24-07375-f003:**
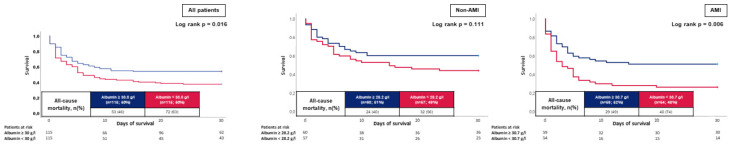
Prognostic impact of albumin on day 1 on the risk of all-cause mortality at 30 days within the entire study cohort (**left** panel), as well as stratified by patients with AMI-related CS (**middle** panel) and non-AMI-related CS (**right** panel).

**Figure 4 ijms-24-07375-f004:**
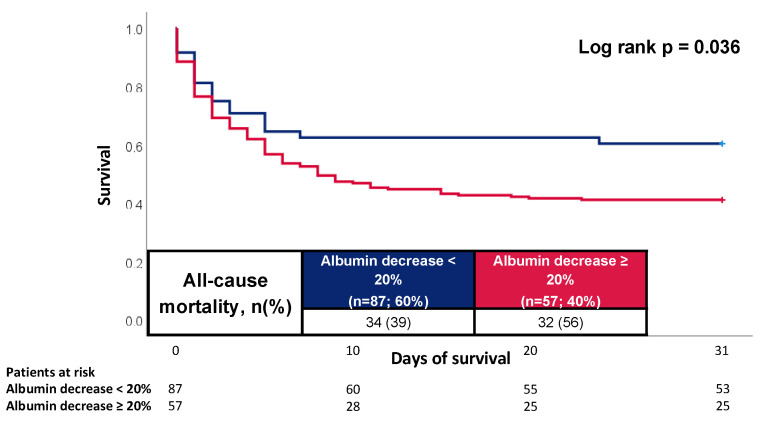
Prognostic impact of albumin decrease from day 1 to day 3 of ICU treatment on the risk of all-cause mortality at 30 days.

**Figure 5 ijms-24-07375-f005:**
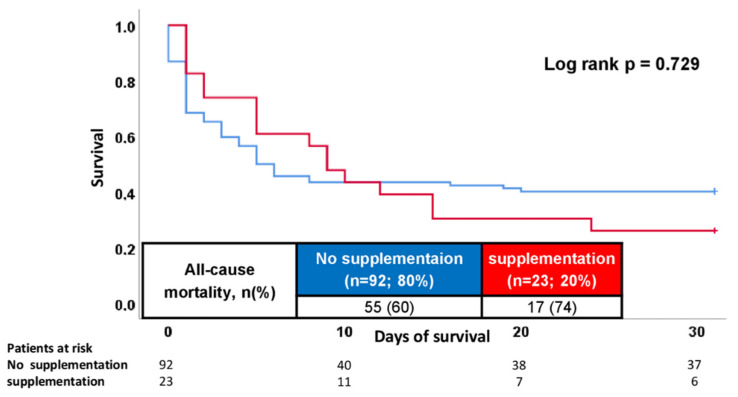
Prognostic impact of albumin supplementation on the risk of all-cause mortality at 30 days in patients with hypalbuminaemia.

**Figure 6 ijms-24-07375-f006:**
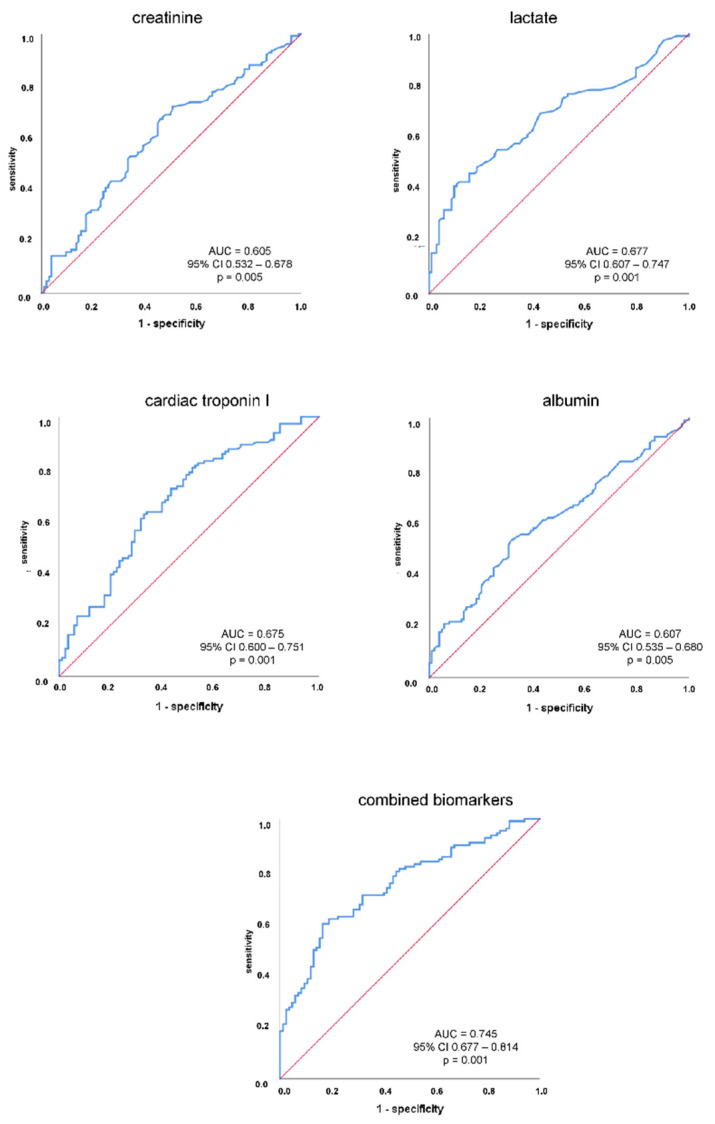
C-statistics for creatinine, lactate, cardiac troponin I, albumin, and the combined use of biomarkers with regard to 30-day all-cause mortality.

**Table 1 ijms-24-07375-t001:** Baseline characteristics.

	All Patients (*n* = 230)		Survivor (*n* = 105)		Non-Survivor (*n* = 125)		*p* Value
Age, median; (IQR)	74	(63–81)	72	(62–80)	74	(64–81)	0.438
Male sex, *n* (%)	143	(62.2)	65	(61.9)	78	(62.4)	0.939
Body mass index, kg/m^2^ (median (IQR))	26.60	(24.20–30.00)	26.30	(24.20–29.40)	26.75	(24.55–30.50)	0.223
Vital signs on admission (median, (IQR))							
Body temperature (°C)	36.0	(35.0–36.6)	36.2	(35.3–36.6)	35.7	(34.5–36.4)	**0.004**
Heart rate (bpm)	88	(71–109)	85	(69–103)	96	(73–112)	**0.017**
Systolic blood pressure (mmHg)	109	(92–127)	112	(94–131)	(103	(89–124)	0.088
Respiratory rate (breaths/min)	20	(17–24)	19	(16–22)	20	(18–25)	0.100
Cardiovascular risk factors, *n* (%)							
Arterial hypertension	166	(72.2)	79	(75.2)	87	(69.6)	0.342
Diabetes mellitus	95	(41.5)	40	(38.5)	55	(44.0)	0.397
Hyperlipidemia	124	(53.9)	59	(56.2)	65	(52.0)	0.525
Smoking	83	(36.2)	37	(35.6)	46	(36.8)	0.848
Prior medical history, *n* (%)							
Coronary artery disease	87	(37.8)	39	(37.1)	48	(38.4)	0.583
Congestive heart failure	82	(35.7)	39	(37.1)	43	(34.4)	0.665
Atrial fibrillation	77	(33.5)	36	(34.3)	41	(32.8)	0.812
Chronic kidney disease	83	(36.1)	40	(380.1)	43	(34.4)	0.561
Stroke	31	(13.5)	18	(17.1)	13	(10.4)	0.136
COPD	42	(18.3)	15	(14.3)	27	(21.6)	0.153
Liver cirrhosis	8	(3.5)	5	(4.8)	3	(2.4)	0.330
Medication on admission, *n* (%)							
ACE-inhibitor	79	(37.3)	38	(36.9)	41	(37.6)	0.914
ARB	36	(16.9)	18	(17.3)	18	(16.5)	0.877
Beta-blocker	114	(53.8)	57	(55.3)	57	(52.3)	0.657
Amiodarone	12	(5.2)	4	(3.8)	8	(6.4)	0.602
ARNI	8	(3.8)	5	(4.9)	3	(2.7)	0.415
Aldosterone antagonist	38	(18.0)	18	(17.6)	20	(18.3)	0.895
Diuretics	97	(45.5)	42	(40.8)	55	(110)	0.177
ASA	62	(27.0)	29	(27.6)	33	(26.4)	0.836
P2Y12-inhibitor	19	(8.3)	7	(6.7)	12	(9.6)	0.421
Statin	100	(46.9)	53	(51.5)	47	(42.7)	0.202
Metformin	26	(11.3)	14	(13.3)	12	(9.6)	0.408
SGLT2-inhibitor	8	(3.5)	3	(2.9)	5	(4.0)	0.638
GLP-1-RA	3	(1.3)	1	(1.0)	2	(1.6)	1.000
DPP-4 inhibitors	4	(1.7)	1	(1.0)	3	(2.4)	1.000

COPD, chronic obstructive pulmonary disease; DPP-4, dipeptidyl peptidase 4; GLP1-RA, Glucagon-like peptide-1 receptor agonist; IQR, interquartile range; LVEF, left ventricular ejection fraction; SGLT2, sodium-glucose cotransporter 2. Level of significance *p* < 0.05. The bold type indicates statistical significance.

**Table 2 ijms-24-07375-t002:** Shock-related data, follow-up data, and endpoints.

	All Patients (*n* = 230)		Survivor (*n* = 105)		Non-Survivor (*n* = 125)		*p* Value
Cause of CS, *n* (%)							**0.002**
Acute myocardial infarction	113	(49.1)	44	(41.9)	69	(55.2)	
STEMI	87	(37.8)	35	(34.3)	52	(44.8)	
NSTEMI	28	(12.2)	9	(8.8)	19	(16.4)	
Arrhythmic	28	(12.2)	21	(20.0)	7	(5.6)	
ADHF	58	(25.2)	24	(22.9)	34	(27.2)	
Pulmonary embolism	12	(5.2)	3	(2.9)	9	(7.2)	
Valvular	10	(4.3)	7	(6.7)	3	(2.4)	
Aortic stenosis	4	(1.7)	3	(2.9)	1	(0.8)	
Sent to valvular surgery	1	(0.4)	1	(1.0)	0	(0)	
Mitral regurgitation	5	(2.2)	3	(2.9)	2	(1.6)	
Tricuspid regurgitation	1	(0.4)	1	(1.0)	0	(0)	
Cardiomyopathy	6	(2.6)	3	(2.9)	3	(2.4)	
Pericardial tamponade	3	(1.3)	3	(2.9)	0	(0.0)	
Onset of CS							
CS on admission, *n* (%)	173	(75.2)	79	(75.2)	94	(75.2)	1.000
CS during hospital stay, *n* (%)	57	(24.8)	26	(24.8)	31	(24.8)	
Days until CS onset, median (IQR)	2.0	(0.0–7.0)	3.5	(1.0–7.3)	1.0	(0.0–7.0)	0.190
Classification of CS, *n* (%)							
Class A	0	(0.0)	0	(0.0)	0	(0.0)	**0.001**
Class B	6	(2.6)	6	(5.7)	0	(0.0)	
Class C	81	(35.2)	49	(46.7)	32	(25.2)	
Class D	16	(7.0)	7	(6.7)	9	(7.2)	
Class E	127	(55.2)	43	(41.0)	84	(67.2)	
Transthoracic echocardiography							
LVEF > 55%, (*n*, %)	22	(9.6)	12	(11.4)	10	(8.0)	
LVEF 54–41%, (*n*, %)	29	(12.6)	19	(18.1)	10	(8.0)	
LVEF 40–30%, (*n*, %)	55	(23.9)	32	(30.5)	23	(18.4)	**0.003**
LVEF < 30%, (*n*, %)	109	(47.4)	37	(35.2)	72	(57.6)	
LVEF not documented, (*n*, %)	15	(6.5)	5	(4.8)	10	(8.0)	
VCI, cm (median, (IQR))	1.9	(1.5–2.2)	1.8	(1.5–2.2)	2.0	(1.6–2.2)	0.264
TAPSE, mm (median, (IQR))	15	(12–19)	17	(12–20)	15	(12–17)	0.159
Coronary angiography at index, *n* (%)	165	(71.7)	80	(76.2)	85	(68.0)	0.188
No evidence of CAD	20	(12.1)	15	(18.8)	5	(5.9)	**0.056**
1-vessel disease	31	(18.8)	16	(20.0)	15	(17.6)	
2-vessel disease	29	(17.6)	14	(17.5)	15	(17.6)	
3-vessel disease	85	(51.5)	35	(43.8)	50	(58.8)	
Affected coronary vessels							
Left main trunc	19	(11.5)	10	(12.5)	9	(10.6)	0.701
Left anterior descending	92	(55.8)	38	(47.5)	54	(63.5)	**0.038**
Right coronary artery	82	(49.7)	39	(48.8)	43	(50.6)	0.813
Left circumflex	80	(48.5)	30	(37.5)	50	(58.8)	**0.006**
Presence of CABG	16	(9.7)	8	(10.0)	8	(9.4)	0.898
Chronic total occlusion	40	(24.2)	13	(16.3)	27	(31.8)	**0.020**
Successful PCI	113	(68.5)	46	(57.5)	67	(78.8)	**0.003**
Failed PCI	3	(1.8)	1	(1.3)	2	(2.4)	1.000
Number of stents, median (IQR)	1.0	(0.0–3.0)	1.0	(0.0–2.0)	2.0	(1.0–3.0)	**0.001**
Stent to CABG, *n* (%)	7	(3.0)	5	(4.7)	2	(1.6)	0.810
Cardiopulmonary resuscitation							
OHCA, *n* (%)	89	(38.7)	33	(31.4)	56	(44.8)	**0.001**
IHCA, *n* (%)	38	(16.5)	10	(9.5)	28	(22.4)	
Shockable rhythm, *n* (%)	160	(70.2)	73	(70.2)	87	(70.2)	0.996
Non-shockable rhythm, *n* (%)	68	(29.8)	31	(29.8)	37	(29,8)	
ROSC, min (median, IQR)	15	(10–29)	10	(5–20)	19	(11–31)	**0.001**
Targeted temperature management, *n* (%)	60	(26.1)	21	(20.2)	39	(31.2)	0.059
Respiratory status							
Mechanical ventilation, *n* (%)	131	(57.7)	48	(45.7)	83	(68.0)	**0.001**
Duration of mechanical ventilation, days, (mean, (IQR))	2	(1–5)	2	(0–6)	2	(1–5)	**0.038**
PaO_2_/FiO_2_ ratio, (median, (IQR))	214	(142–357)	234	(149–367)	213	(120–343)	0.367
PaO_2_, mmHg (median, (IQR))	103	(79–166)	103	(79–163)	106	(78–171)	0.850
ICU-related treatment							
Norepinephrine dose, µg/kg/min (median, (IQR))	0.1	(0.0–0.3)	0.1	(0.0–0.1)	0.2	(0.1–0.6)	**0.001**
Mechanical circulatory support device, *n* (%)	22	(9.6)	2	(1.9)	20	(16.0)	**0.001**
Dobutamine, *n* (%)	41	(17.8)	13	(12.4)	28	(22.4)	0.057
Levosimendan, *n* (%)	60	(26.1)	17	(16.2)	43	(34.4)	**0.002**
Renal replacement therapy, *n* (%)	67	(29.1)	15	(14.3)	52	(41.6)	**0.001**
Fluid balance at 24 h, mL/d, median (IQR)	953	(40–2256)	597	(−243–1247)	1562	(343–3501)	**0.001**
Albumin supplementation, *n* (%)	23	(10.0)	8	(5.7)	17	(13.6)	**0.050**
Albumin supplementation dose during ICU stay, g, median (IQR)	80	(40–120)	110	(80–205)	80	(40–120)	0.155
Infection, *n* (%)	120	(52.2)	57	(54.3)	63	(50.4)	0.557
Pulmonary, *n* (%)	93	(40.4)	42	(40.0)	51	(41.6)	0.806
Urogenital, *n* (%)	18	(7.8)	12	(11.4)	6	(4.8)	0.062
Other, *n* (%)	9	(3.9)	3	(2.9)	6	(4.8)	0.771
Bleeding, *n* (%)	23	(10.0)	12	(12.4)	10	(8.0)	0.270
Baseline laboratory values, (median, (IQR))							
pH	7.29	(7.21–7.37)	7.32	(7.26–7.37)	7.28	(7.16–7.36)	**0.003**
Lactate (mmol/L)	3.3	(1.7–6.8)	2.6	(1.6–4.2)	4.5	(2.5–10.2)	**0.001**
Sodium (mmol/L)	138	(136–141)	138	(136–140)	138	(136–141)	0.321
Potassium (mmol/L)	4.3	(3.8–4.9)	4.2	(3.7–4.8)	4.4	(3.9–5.0)	0.222
Creatinine (mg/dL)	1.48	(1.13–2.17)	1.32	(1.07–1.83)	1.60	(1.23–2.39)	**0.006**
Hemoglobin (g/dL)	12.3	(10.3–14.0)	12.2	(10.1–14.2)	12.3	(10.5–13.8)	0.638
WBC (10^6^/mL)	15.09	(10.66–19.20)	13.10	(9.70–17.66)	16.12	(12.34–19.96)	**0.001**
Platelets (10^6^/mL)	225	(171–275)	225	(163–287)	225	(177–267)	0.892
INR	1.17	(1.08–1.39)	1.13	(1.05–1.33)	1.20	(1.10–1.47)	**0.003**
D-dimer (mg/L)	10.23	(2.58–32.00)	6.40	(2.09–16.55)	18.21	(3.64–32.00)	**0.008**
AST (U/L)	129	(47–408)	96	(36–191)	189	(64–545)	**0.002**
ALT (U/L)	77	(31–200)	52	(28–124)	96	(35–339)	**0.006**
Bilirubin (mg/dL)	0.62	(0.43–0.98)	0.60	(0.41–0.93)	0.64	(0.47–1.01)	0.227
Albumin (g/L)	30.0	(25.5–33.9)	30.7	(27.6–34.4)	28.7	(23.8–33.0)	**0.005**
Troponin I (µg/L)	0.841	(0.912–7.678)	0.348	(0.090–2.561)	1.952	(0.424–12.434)	**0.001**
NT-pro BNP (pg/mL)	4063	(681–13606)	3621	(466–11732)	4122	(1098–13629)	0.287
Procalcitonin (ng/mL)	0.28	(0.12–0.96)	0.29	(0.07–0.77)	0.28	(0,18–1.66)	0.474
CRP (mg/L)	12	(4–41)	6	(4–42)	15	(4–41)	0.362
Follow-up data, *n* (%)							
ICU time, days (median, (IQR))	4	(2–8)	4	(2–10)	3	(2–6)	**0.001**

ADHF, acute decompensated heart failure; ALT, alanine aminotransferase; AST, aspartate aminotransferase; CABG, coronary artery bypass grafting; CAD, coronary artery disease; CRP, C-reactive Protein; ICU, intensive care unit; IHCA, in-hospital cardiac arrest; INR, international normalized ratio; IQR, interquartile range; (N)STEMI, (non-)ST-segment myocardial infarction; NT-pro BNP, aminoterminal pro-B-type natriuretic peptide; OHCA, out-of-hospital cardiac arrest; PCI, percutaneous coronary intervention; ROSC, return of spontaneous circulation; TAPSE, tricuspid annular plane systolic excursion; VCI, Vena cava inferior; WBC, white blood cells. Level of significance *p* < 0.05. The bold type indicates statistical significance.

**Table 3 ijms-24-07375-t003:** Univariate correlations of albumin levels with laboratory and clinical parameters in all patients (*n* = 230) at day 1.

	*R*	*p* Value
Age	−0.147	**0.026**
BMI	−0.038	0.576
Body temperature	0.006	0.935
WBC	−0.017	0.800
Platelets	0.101	0.127
Bilirubin	−0.153	0.063
Creatinine	−0.285	**0.001**
CRP	−0.433	**0.001**
PCT	−0.392	**0.001**
cTNI	−0.061	0.392
NT-pro BNP	−0.596	**0.001**
LVEF	−0.103	0.120
PaO_2_/FiO_2_ ratio	0.127	0.074
Mechanical ventilation days	0.038	0.562
Fluid balance	−0.201	**0.003**
SOFA score	−0.409	**0.001**
Acute Physiology score	−0.339	**0.001**
APACHE II score	−0.413	**0.001**
Intensive care days	0.103	0.121

APACHE II; acute physiology and chronic health evaluation II; BMI, body mass index; CRP, C-reactive protein; cTNI, cardiac troponin I; LVEF, left ventricular ejection fraction; MAP, mean arterial pressure; NT-pro BNP, N-terminal pro-B-type natriuretic peptide; SOFA, sepsis-related organ failure assessment; PCT, procalcitonin; WBC, white blood cell count. Level of significance *p* < 0.05. The bold type indicates statistical significance.

**Table 4 ijms-24-07375-t004:** Prognostic performance of the albumin level on days 1, 2, 3, 4, and 8, analyzed as the area under the curve (95% CI).

	Albumin
Day 1	0.607 (0.535–0.680); ***p* = 0.005**
Day 2	0.635 (0.554–0.716); ***p* = 0.002**
Day 3	0.681 (0.594–0.768); ***p* = 0.001**
Day 4	0.683 (0.587–0.779); ***p* = 0.001**
Day 8	0.753 (0.638–0.868); ***p* = 0.001**

Level of significance *p* < 0.05. The bold type indicates statistical significance.

**Table 5 ijms-24-07375-t005:** Uni- and multivariable Cox regression analyses with regard to 30-day all-cause mortality within the entire study cohort.

Variables	Univariable			Multivariable		
	HR	95% CI	*p* Value	HR	95% CI	*p* Value
Age	1.006	0.993–1.020	0.364	1.008	0.993–1.023	0.287
Sex	1.020	0.710–1.465	0.916	0.837	0.556–1.259	0.393
BMI	1.018	0.983–1.054	0.317	1.007	0.967–1.048	0.732
Systolic BP	0.993	0.987–1.000	**0.044**	0.996	0.989–1.003	0.237
Respiratory rate	1.025	0.995–1.055	0.105	1.030	1.000–1.061	0.052
Hb	0.979	0.913–1.049	0.545	0.991	0.912–1.078	0.837
WBC count	1.039	1.012–1.067	**0.005**	1.054	1.022-.087	**0.001**
Platelet count	0.999	0.998–1.001	0.586	0.999	0.996–1.001	0.283
Levosimendan	1.005	1.001–1.008	**0.017**	1.005	1.001–1.009	**0.020**
Fluid balance	2.087	1.265–3.441	**0.004**	1.831	1.093–3.066	**0.021**
Albumin < 30.0 g/L	1.517	1.063–2.164	**0.021**	1.875	1.109–3.170	**0.019**

BMI, body mass index; BP, blood pressure; Hb, hemoglobin; WBC, white blood cell count. Level of significance *p* < 0.05. The bold type indicates statistical significance.

**Table 6 ijms-24-07375-t006:** Prognostic impact of albumin decrease within uni- and multivariable Cox regression analyses.

Variables	Univariable			Multivariable		
	HR	95% CI	*p* Value	HR	95% CI	*p* Value
Age	1.006	0.993–1.020	0.364	1.003	0.985–1.023	0.720
Sex	1.020	0.710–1.465	0.916	0.693	0.396–1.215	0.201
BMI	1.018	0.983–1.054	0.317	0.973	0.919–1.030	0.339
Systolic BP	0.993	0.987–1.000	**0.044**	0.994	0.985–1.003	0.188
Respiratory rate	1.025	0.995–1.055	0.105	1.023	0.985–1.003	0.237
Hb	0.979	0.913–1.049	0.545	0.951	0.854–1.058	0.355
WBC	1.039	1.012–1.067	**0.005**	1.037	0.995–1.082	0.087
Platelet count	0.999	0.998–1.001	0.586	0.999	0.996–1.001	0.341
Albumin decrease ≥ 20%	1.645	1.014–2.669	**0.044**	1.737	1.029–2.933	**0.039**

BMI, body mass index; BP, blood pressure; Hb, hemoglobin; WBC, white blood cell count. Level of significance *p* < 0.05. The bold type indicates statistical significance.

## Data Availability

The datasets used and/or analyzed during the current study are available from the corresponding author upon reasonable request.

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
