# Peer review of "Does Albumin Predict the Risk of Mortality in Patients with Cardiogenic Shock?"

_ijms, 2023, doi:10.3390/ijms24087375_

Round 1

Reviewer 1 Report

This study investigates the prognostic impact of albumin levels in patients with cardiogenic shock. The topic of this paper is very meaningful. It has also done a lot of difficult data collection work and obtained some valuable results. However, there are some problems in the following aspects:

1. The sample size is very small, and the representativeness and reliability of the results may have great problems.

2. The specific test method and instrument information of each index of laboratory test should be introduced in the method section.

3. Is there any corresponding quality control in the process of data collection? The method description of specific control should be supplemented in the method section.

4. From the AUC value, the albumin level is not of high predictive value for CA. Have you considered the joint predictive analysis of other indicators?

5. Only observing the 30-day survival  has great limitations. Although there is no statistical difference between the survival and death populations in the comparison of disease history and medication history, there may be a certain selection bias, which may deviate from the actual situation. It is suggested to add some sensitivity analysis or subgroup analysis to further verify the reliability of the results.

Author Response

We would kindly like to thank the reviewer for the comments regarding our study.

Please find attached a detailed point-to-point response.

Reviewer 2 Report

Please find my comments in the document attached. 

Author Response

(The authors gave the same response as above.)

Round 2

Reviewer 1 Report

None